# A Systematic Review of Associations between Energy Use, Fuel Poverty, Energy Efficiency Improvements and Health

**DOI:** 10.3390/ijerph19127393

**Published:** 2022-06-16

**Authors:** Chengju Wang, Juan Wang, Dan Norbäck

**Affiliations:** Department of Medical Sciences, Occupational and Environmental Medicine, Uppsala University, 75185 Uppsala, Sweden; dan.norback@medsci.uu.se

**Keywords:** health, asthma, respiratory, indoor air quality, built environment, energy use, energy efficiency buildings, green buildings

## Abstract

Energy use in buildings can influence the indoor environment. Studies on green buildings, energy saving measures, energy use, fuel poverty, and ventilation have been reviewed, following the guidelines of the Preferred Reporting Items for Systematic Reviews and Meta-Analyses (PRISMA) statement. The database PubMed was searched for articles published up to 1 October 2020. In total, 68 relevant peer-reviewed epidemiological or exposure studies on radon, biological agents, and chemicals were included. The main aim was to assess current knowledge on how energy saving measures and energy use can influence health. The included studies concluded that buildings classified as green buildings can improve health. More efficient heating and increased thermal insulation can improve health in homes experiencing fuel poverty. However, energy-saving measures in airtight buildings and thermal insulation without installation of mechanical ventilation can impair health. Energy efficiency retrofits can increase indoor radon which can cause lung cancer. Installation of a mechanical ventilation systems can solve many of the negative effects linked to airtight buildings and energy efficiency retrofits. However, higher ventilation flow can increase the indoor exposure to outdoor air pollutants in areas with high levels of outdoor air pollution. Finally, future research needs concerning energy aspects of buildings and health were identified.

## 1. Introduction

In modern society, people spend more than 90% of their time in indoor environments, and most of that time is spent at home [1]. Energy is needed to heat or cool buildings, and energy use in buildings is an important issue in contemporary society [2]. The climate change issue, linked to increased greenhouse gases emissions from coal, oil, or gas combustion, has increased the demand to save energy in buildings in different parts of the world [3]. Because of this demand, different measures have been applied to increase energy efficiency in buildings in order to create a sustainable built environment which combines a healthy and energy-efficient indoor environment [4].

There are three main principles of energy efficiency improvements in buildings: reduced energy use, reduced heat transfer, and reduced air leakage [5]. Reduced energy use can reduce emissions and fuel cost, thus reducing exposure to emissions [6]. Furthermore, reduced heat transfer can increase indoor temperature and reduce relative humidity and risk of mould [7]. In contrast, reduced air leakage can increase relative humidity and risk of mould [8]. In practice, common energy saving measures in buildings include increased thermal insulation, installation of central heating or space heating, draught proofing or installation of heat recovery systems [9]. Since energy use in buildings is a complex issue, scientists from many disciplines, as well as stake holders, government officers, and other decision makers need to work together to make updated energy policies [10].

In recent years, there has been an increase of energy-related labelling of buildings, e.g., low energy buildings, zero energy buildings, green buildings, and healthy buildings [11,12,13]. Green building rating systems have been widely used globally for many years [13]. In the USA, they have created Leadership in Energy and Environmental-Design (LEED) credits to assess green buildings [12]. Other existing green rating systems include BREEAM, CASBEE, Green Star, Enterprise Green Communities, RELi, SITES, Fitwel, Living Building Challenge (LBC), and WELL [11].

However, it should be realized that extreme cold in homes in winter can increase cold-related mortality or morbidity rates [14]. In the UK, fuel poverty is definite as people don’t have enough money to heat their home in winter to maintain an acceptable temperature [15]. However, there is also a cost for cooling their homes in extreme heat situations which some people cannot afford [16].

This systematic review included all types of health aspects of energy use, energy saving, and energy efficiency in buildings. The main aim was to summarize the current knowledge on the health impacts of energy saving measures and energy use. The second aim was to collect knowledge on the indoor environment effects of energy-saving measures and energy use. The third aim was to gather knowledge on types of energy saving or energy use that should be promoted from a health perspective.

## 2. Methods

The guidelines of the Preferred Reporting Items for Systematic Reviews and Meta-Analyses (PRISMA) statement were followed to perform this systematic review [17]. In October 2020, a systematic literature search in PubMed covering articles up to 1 October 2020 was performed. There were ten medical search terms: morbidity, mortality, respiratory, lung function, asthma, rhinitis, eczema, dermatitis, sick building syndrome, building related illness. These medical search terms were combined (any of the ten search terms). In addition, there were eight energy and building related search terms: energy saving, energy use building, energy efficiency building, energy consumption building, energy efficient building, low energy building, energy retrofit, green building. These building and energy-related medical search terms were combined (any of the eight search terms). Then a systematic database search combining any of the ten medical search terms with any of the eight energy and building related search terms was performed. Any medical search term means OR between each search term. Any energy or building related search term means OR between each search term. Combined means AND between the two groups of search terms.

In total, 5776 records were identified from the database searching. Those records were sent to EndNote citation manager for collecting, storing, and organizing. In this reference management software, three reference groups (duplicated group, included group, excluded group) were created. First, 806 duplicated records were removed and added into duplicated group before screening by using the function of EndNote. Then the titles and the abstracts of 4970 articles were screened, to identify articles relevant to the topic of this literature review.

The following three selection criteria were used to include studies in this review:The articles should have studied associations between energy aspects in buildings and health;The articles should be written in English;The articles should not be keynotes, opinions, commentaries, reviews, or modelling studies.

In total, 4882 irrelevant articles were removed. After removal of irrelevant articles, 88 relevant articles were identified. As a next step, keynotes, opinions, commentaries, review articles and modelling articles were removed. Finally, 68 relevant field studies were included in this review, of which 45 were health studies and 23 studies had measured exposure in relation to energy aspects in buildings without investigating health associations. The PRISMA flow diagram of the literature research is shown in Figure 1.

For each health study, study characteristics on author, year, country, energy aspects, type of study, type of buildings, type of health variables, number of buildings, number of subjects, and main results were extracted. For each exposure study, study characteristics on author, year, country, energy aspects, type of study, type of buildings, measured exposure, changes of measured exposure, number of households, or buildings and main results were extracted. In addition, within the health studies, articles with positive and negative health associations were grouped.

In order to further organize the structure of tables, a thematic classification was made. The studies were divided into four categories: exposure studies, green building health studies, fuel poverty health studies, other energy-related health studies. The exposure studies were divided into three exposure groups, including exposure to radon, exposure to biological agents (mould, bacteria, and house dust mites) and exposure to chemicals. The fuel poverty health studies were divided into three health aspects, including respiratory symptoms, general and mental health, and studies on mortality. Other energy-related health studies were divided into cross-sectional heath studies, longitudinal studies, and intervention health studies according to the study design. Details on those thematic tables can be seen in the Appendix A.

The entire process above involved at least two authors to conduct searching to gathering, screening, analyzing, and extracting.

## 3. Results

### 3.1. Exposure Studies

In Table 1, associations between energy-related building factors and indoor pollutants among the 23 included exposure studies are summarized. These included 19 studies conducted in Europe, 3 studies conducted in USA, and 1 study conducted in China. Except for one school study [18], 22 exposure studies were conducted in residential buildings (Appendix A).

#### 3.1.1. Radon

There were 11 exposure studies on radon [19,20,21,22,23,24,25,26,27,28,29] (Table A1). Of these, 9 studies reported that energy efficiency thermal retrofitting in homes increased radon concentration [19,20,21,22,23,24,25,27,29]. Of these 9 studies, 3 combined thermal insulation with additional air sealing methods in windows [21,22,27]. There were 6 studies of the 9, in five countries, which reported average radon concentrations above 100 Bq/m^3^ in rooms [19,20,22,24,25,27]. However, three studies of 11 demonstrated that energy efficiency retrofitting in homes with installation of mechanical ventilation or other measures can reduce radon concentration [26,27,28]. Other measures included installation of ground covers [26,27] and sub-slab or sump depressurization systems [26].

#### 3.1.2. Biological Agents

There were 8 exposure studies on biological agents [28,29,30,31,32,33,34,35] (Table A2). One study reported that installation of insulated windows and central heating systems increased the concentration of the house dust mites and mould [30]. Another study showed that fuel poverty can increase indoor dampness and mould, regardless of the use of extractor fans [31]. The negative effects may be caused by reduced ventilation [30] and ineffective heating [31]. However, 6 studies of 8 found that energy efficiency improvement in homes with improved ventilation can reduce indoor exposure to mould [28,29,32,34], bacteria [29] and house dust mites [33].

#### 3.1.3. Chemical Substances and Particles

There were 9 exposure studies on chemical substances and particles (Table A3). 4 studies demonstrated that home energy efficiency retrofit can increase indoor air concentrations of certain volatile organic compounds [29,36,38,39] and carbon dioxide levels (CO_2_) [39]. CO_2_ is an indicator of ventilation flow rate. Those volatile organic compounds included formaldehyde [38,39], aromatics [39], alkanes [39] and alpha-pinene [36], hexaldehyde [36], as well as benzene, toluene, ethyl benzene, and xylene (BTEX) [29]. Alpha-pinene and hexaldehyde could be caused by the use of wood or wood-based products for construction and insulation [36]. However, some studies reported that home energy efficiency improvement combined with mechanical ventilation system can reduce aldehydes [28], formaldehyde [29], total volatile organic compounds (TVOC) [28], CO_2_ [18,28,37], carbon monoxide (CO) [27], and black carbon level [38]. One study found that an energy intervention replacing low-polluting semigasifier cooking stoves in rural buildings was associated with decreased exposures to 2.5 (PM_2.5_) particulate matter and black carbon in winter but higher exposure in summer. The negative effect could be caused by increased use of the cooking stove [40].

### 3.2. Health Studies

In Table 2, associations between one kind of fuel poverty, improved ventilation, and energy efficiency improvements and health are summarized. There were 28 studies which were conducted in Europe, 10 studies conducted in the USA, and 7 in other countries, including New Zealand (*n* = 3), Japan (*n* = 2), Canada (*n* = 1), and India, (*n* = 1). Except for three office [41,42,43] and three school studies [44,45,46], 39 studies were performed in residential buildings (Appendix A).

#### 3.2.1. Green Building Health Studies

The green building health studies were conducted in United States (*n* = 5), Canada (*n* = 1), and India (*n* = 1). They were performed in two offices [41,42], two schools [44,45] and three residential buildings (Table A4). Some studies demonstrated that green buildings can reduce self-reported asthma [41,47,48], non-asthmatic respiratory symptoms [41,48], and improve general health [44,45,48,49] and mental health [41,49] as well as performance [41,44,45] and satisfaction [44,45]. One study found no significant association between green buildings and sick building syndrome symptoms (SBS) [42]. Sick building syndrome symptoms include nonspecific symptoms from eyes, skin, upper airways, headache, and fatigue [1].

#### 3.2.2. Fuel Poverty Studies

The fuel poverty studies were conducted in the United Kingdom (*n* = 10), the USA (*n* = 3), New Zealand (*n* = 3), Spain (*n* = 2), Japan (*n* = 1), and multiple countries (*n* = 1). All of the 20 studies were conducted in residential buildings (Table A5, Table A6 and Table A7). Some studies reported that fuel poverty in low-income homes can increase asthma [52] and respiratory symptoms [50,51] and reduce general health [57] and mental health [57]. Furthermore, low indoor air temperature in low-income homes can increase blood pressure [64,67] and hypertension [67] (linked to cold-related mortality). Besides, lack of insulation [65] and heating systems [68] in low-income homes can increase cold-related mortality. However, one study showed that wearable telemetry (a thermometer with a low-temperature alarm) can raise awareness of the health effects of cold living environments among people living in fuel poverty (linked to psychosocial outcomes) [63].

There were another some studies on the effects of improved ventilation or energy efficiency improvements in low-income homes and health. First, they found that high ventilation rates in low-income urban homes may increase chronic cough, asthma, and asthma-like symptoms, probably caused by infiltration of outdoor air pollutants [54]. However, high infiltration rates in low-income, urban, non-smoking homes can improve lung health [85]. Second, they demonstrated that installation of cavity wall insulation in social housing without installation of mechanical ventilation can reduce general health outcomes and social outcomes [53]. Energy efficient façade insulation retrofits in public housing can reduce cold-related mortality in women, but can increase cold-related mortality in men. The reason for the gender difference is unclear [66]. However, energy efficiency improvements in low-income homes can improve respiratory symptoms [53,55,56], general health [53,55,58,59,60] and mental health [53,58] as well as psychosocial outcomes [53,61,62], well-being [55,59,61,62], and sleep [58].

#### 3.2.3. Cross-Sectional Health Studies

The cross-sectional health studies were conducted in Sweden (*n* = 4), the United Kingdom (*n* = 2), Norway (*n* = 1), and Germany (*n* = 1). Expect for one school study [46], seven studies were performed in residential buildings (Table A8). Some studies investigated the association between ventilation and health. They reported that higher ventilation rate in homes were associated with less asthma symptoms [72,73]. Furthermore, in multi-family buildings, lack of a mechanical ventilation system was associated with increased prevalence of SBS-related symptoms [69]. Further, buildings with balanced ventilation systems (supply/exhaust ventilation) had a higher prevalence of doctor diagnosed allergies, as compared to buildings with exhaust ventilation only [71].

There were some other investigative studies on the health impacts of energy efficiency in buildings. First, they found that air tightness [69,74] and use of direct electric radiators [69] in residential buildings were associated with increased prevalence of SBS-related symptoms. However, higher insulation level in buildings was associated with less SBS symptoms [70]. Second, buildings using more energy for heating were associated with lower rates of pollen allergies and eczema [71]. Energy efficiency improvements by boiler replacements in homes were associated with less admission rates for asthma and chronic obstructive pulmonary disease (COPD) [73]. Third, lower air temperature in buildings at a university campus was associated with less tear film stability [46]. Higher thermal variety (linked to lower domestic demand temperatures) was associated with fewer morbidities related to cold mortality [75].

#### 3.2.4. Longitudinal Health Studies

A longitudinal study from Austria found that energy efficient buildings combined with installation of mechanical ventilation can improve general health and mental health but increase dry eye symptoms, as compared to conventional buildings with natural ventilation only [76] (Table A9).

#### 3.2.5. Intervention Health Studies

Intervention health studies were conducted in the United Kingdom (*n* = 3), the United States (*n* = 2), Japan (*n* = 1), Sweden (*n* = 1), Denmark (*n* = 1), and multiple countries (*n* = 1). Except for one office study [43], eight studies were performed in residential buildings (Table A10). Some studies reported that energy efficiency intervention in homes can improve asthma [77,78], respiratory symptoms [77,78,79,81], sinusitis [80], general health [80,83], satisfaction [80,81], and reduce blood pressure [84]. Furthermore, an improved mechanical ventilation rate in office buildings can improve SBS symptoms, productivity, and perceived indoor air quality [43]. In addition, energy saving by reducing ventilation flow to below 0.5 air change rate (ACH) could impair perceived air quality but did not influence SBS [82].

#### 3.2.6. Energy Factors and Health

In Table 3, data on associations between energy factors and any health outcomes among all 45 selected health studies were summarized. Thermal issues, including fuel poverty or low indoor air temperature, were not included in this table. Most studies showed beneficial effects of energy saving.

## 4. Discussion

To our knowledge, this review is the first systematic review on associations between different energy aspects of buildings and health. A meta-analysis could not be performed, since there were few articles covering the same energy aspect and the same health variable. However, the current knowledge level and knowledge gaps on the health effects of green buildings, fuel poverty, and energy use as well as energy efficiency improvements in buildings was able to be summarized or described.

In this review, there were three important issues related to exposure studies. Firstly, radon concentration in six studies was above 100 Bq/m^3^ in mean or in rooms [19,20,22,24,25,27]. In one review with meta-analysis on the risk of radon, the action level of radon for never-smokers and ever-smokers was recommended at 100 Bq/m^3^ of World Health Organization. They reported that radon exposure is the strongest risk factor for lung cancer for never-smokers [86]. Thus, special concern should be taken around radon exposure when performing home energy efficiency retrofits. In order to reduce radon levels in home energy-efficiency retrofits, installation of ground covers and sub-slab or sump depressurization systems as well as mechanical ventilation could be undertaken. One main source of indoor radon is radon from the ground. It should be ensured that the transmission of radon from the ground into buildings is minimized, especially for buildings in regions with primary geological layers in the underground. Another source of indoor radon is building materials, although it is not the main source. It is highly recommended that the building material for home retrofits works should meet the standards of green buildings. Secondly, installation of insulated windows and central heating systems can increase the indoor concentrations of mould [30]. The health risk of mould had been assessed in a previous review [8]. In many countries, mould and dampness caused by critical thermal bridges is a reason why energy efficiency interventions were performed [87]. Thus, it is important to consider thermal bridges as a cause of indoor mould growth after improving insulation in buildings. Thirdly, home energy efficiency retrofits can increase benzene, toluene, ethyl benzene, and xylene (BTEX) in indoor air [29]. In one previous review, the negative health effects of indoor BTEX had been reported [88]. Thus, it is important to use low-emissions building materials in energy efficiency retrofits.

Moreover, there were four important issues related to health studies.

Firstly, there were negative health effects in buildings with thermal insulation without installation of mechanical ventilation. In most cases, thermal insulation can reduce heat transfer, which will increase indoor temperature and reduce relative humidity and risk of mould. However, since many energy efficiency improvement methods can lead to reduced ventilation rates or air tightness, special concern should be taken to compensate for the reduced natural ventilation rate when working with home energy efficiency improvements. Thus, energy efficiency methods combined with improved ventilation or design should be promoted in airtight homes. In addition, the issue of thermal bridges and mould growth was seldom mentioned in the health studies.

Secondly, there were two negative associations between improved ventilation rate and health. In a fuel poverty study, high ventilation rates in low-income urban homes may increase chronic cough, asthma, and asthma-like symptoms [54]. This could be due to increased infiltration of outdoor air pollutants. Although this knowledge may be well known, the level of outdoor air pollutants had not been evaluated by the current intervention programs of low-income homes we found. In a cross-sectional health study, buildings with balanced ventilation systems (supply/exhaust ventilation) had a higher prevalence of doctor diagnosed allergies, as compared to buildings with exhaust ventilation only [71]. This may be caused by lack of a correct replacement of dirty filters in balanced mechanical ventilation systems. Thus, this knowledge should be addressed to residents in homes with energy efficiency improvements combined with balanced ventilation systems.

Thirdly, four fuel poverty health studies on cold mortality were performed in a longitudinal study design. This means that the cold-mortality effect of fuel poverty has been well known. Thus, fuel poverty behavior should be considered in interventions since it is often linked to reduced ventilation rate and ineffective heating. Except for winter fuel payment and energy intervention policy, wearable telemetry may be a good choice of solution in cold homes [63]. This is because wearable telemetry can increase the occupant’s awareness of cold. However, all those studies were based on cold climates. In hot climate zones, there is a need to conduct similar research in low-income homes.

Fourthly, 4 green buildings health studies were conducted in a longitudinal study design. This means that long-term health effects of green buildings were assessed in the USA. However, those green buildings were assessed by LEED credits of the USA standard. Although there are existing green rating systems in different countries, energy efficiency improvements combined with correct ventilation and renewable energy use have been emphasized in most green rating systems.

This literature review has a number of strengths. The main focus was on epidemiological studies, including intervention studies, cross-sectional studies, and longitudinal studies. However, exposure studies without any reported health data or health associations were also included if they were identified in this literature search. For each included study, the country of the study, type of study, type of buildings, number of buildings and number of subjects were noted in the review. In exposure studies, extra information on the changes of concentrations of major pollutants was collected. In studies with unexpected results or negative impacts of energy use and energy saving, explanations of the results reported by the authors were included.

The studies included in this review had some limitations in their study design. One major limitation was that none of the studies had studied health effects of energy efficiency improvement by the installation of heat recovery to existing mechanical ventilation systems. This may be because many studies had not separated it from combined energy efficiency measures. However, installation of heat recovery to mechanical ventilation systems is a major method nowadays to save energy use and there is a need to assess its health benefits, especially in airtight homes. The second limitation is that many of the intervention studies were based on more than two energy saving improvements. Thus, it is not possible to draw clear conclusions on the health effects of single energy efficiency improvement measures. The third limitation is that there were few prospective health studies on long-term health effects of energy efficiency improvements and energy use. However, many prospective health studies on green buildings and fuel poverty were found. The fourth limitation is that most studies were on residential buildings. Only three studies were on office buildings and only four studies were on school or university buildings.

## 5. Conclusions

Energy efficiency improvements and green building can have positive effects on asthma, respiratory symptoms, mental health, and general health as well as on performance and satisfaction. Home energy efficiency improvement with mechanical ventilation system can reduce radon, mould, bacteria, and house dust mites, TVOC, CO_2_, CO, and black carbon levels as well as some volatile organic compounds. More efficient heating and increased thermal insulation can have positive health impacts in fuel-poverty homes. However, energy savings in airtight buildings and thermal insulation without the installation of mechanical ventilation can impair health. Moreover, health risks linked to energy efficiency retrofits exists. Installation of mechanical ventilation can solve many of the negative effects linked to airtight buildings and energy efficiency retrofits.

For future energy efficiency intervention or retrofit studies, measures of radon and BTEX and other chemicals, as well as levels of thermal bridge and outdoor air pollutants may be needed. In addition, it is important to replace dirty filters in balanced mechanical ventilation systems.

Furthermore, future research needs on this topic were identified. Firstly, the intervention study should measure how much energy they save after energy efficiency measures. Secondly, more studies are needed on the health aspects of energy efficiency improvement by the installation of heat recovery to mechanical ventilation system. Thirdly, future studies should focus on evaluating health effects of single energy efficiency improvement measures, rather than a combination of measures. Fourthly, more prospective health studies on long-term health effects of energy efficiency improvements or energy use are needed. Fifthly, future studies should include offices, schools, and hospital buildings, and should cover different climate zones in the world.

## Figures and Tables

**Figure 1 ijerph-19-07393-f001:**
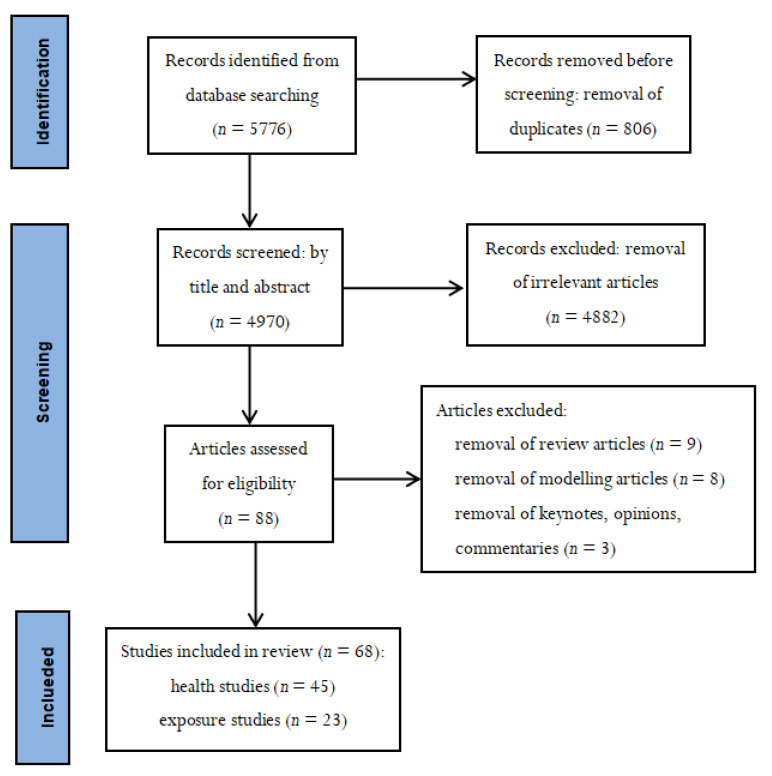
PRISMA flow diagram of literature research.

**Table 1 ijerph-19-07393-t001:** Associations between energy-related building factors and pollutants among the 23 included exposure studies.

No.	References	Country	Pollutant Groups	Improved Ventilation	Thermal Retrofit	Draught Proofing	Green Retrofits	Fuel Poverty	Energy Carrier
1	Collignan et al. 2016 [19]	France	Radon						
2	Symonds et al. 2019 [20]	United Kingdom	Radon		↑				
3	Meyer et al. 2019 [21]	Germany	Radon		↑	↑			
4	Pressyanov et al. 2015 [22]	Bulgaria	Radon		↑	↑			
5	Vasilyev et al. 2017 [23]	Russia	Radon		↑				
6	Yarmoshenko et al. 2014 [24]	Russia	Radon		↑				
7	Vasilyev et al. 2015 [25]	Russia	Radon		↑				
8	Burghele et al. 2020 [26]	Romania	Radon	↓					
9	Pigg et al. 2018 [27]	United States	Radon ^1^, Chemicals ^3^	↓	↑^1^, ↓^3^	↑^1^, ↓^3^			
10	Wallner et al. 2015 [28]	Austria	Radon ^1^, Biological agents ^2^, Chemicals ^3^	↓^1^, ↓^2^, ↓^3^					
11	Du et al. 2019 [29]	Finland Lithuania	Radon ^1^, Biological agents ^2^, Chemicals^3^	↓^2^, ↑↓^3^	↑^1^				
12	Hirsch et al. 2020 [30]	Germany	biological agents		↑				
13	Sharpe et al. 2015 [31]	United Kingdom	biological agents					↑	
14	Sharpe et al. 2016 [32]	United Kingdom	biological agents	↓					
15	Spertini et al. 2010 [33]	Switzerland	biological agents	↓					
16	Niculita-Hirzel et al. 2000 [34]	Switzerland	biological agents	↓					
17	Coombs et al. 2018 [35]	United States	biological agents				0		
18	Derbez et al. 2018 [36]	France	Chemicals		↑				
19	Leivo et al. 2018 [37]	Finland Lithuania	Chemicals	↓	↑				
20	Coombs et al. 2016 [38]	United States	Chemicals				↑↓		
21	Yang et al. 2020 [39]	Switzerland	Chemicals		↑				
22	Verriele et al. 2016 [18]	France	Chemicals	↓					
23	Baumgartner et al. 2019 [40]	China	Chemicals						↑↓

↑ means increase, ↓ means decrease, ↑↓ means mixed results, 0 means no associations. ^1^ represents radon, ^2^ represents biological agents, ^3^ represents chemicals.

**Table 2 ijerph-19-07393-t002:** Associations between one kind of fuel poverty, improved ventilation, and energy efficiency improvements and health in all 45 selected health studies.

No.	Reference	Thematic Group	Respiratory Health		General Health	Mental Health	Performance	Satisfaction	Cold-Related Mortality	SBS Symptoms
			Asthma	Other Respiratory Illnesses						
1	Garland et al. 2013 [47]	Green Buildings	+							
2	Singh et al. 2010 [41]	Green Buildings	+	+		+	+			
3	Breysse et al. 2011 [48]	Green Buildings	+	+	+					
4	Breysse et al. 2015 [49]	Green Buildings			+	+				
5	Hedge et al. 2013 [44]	Green Buildings			+		+	+		
6	Hedge et al. 2014 [45]	Green Buildings			+		+	+		
7	Gawande et al. 2020 [42]	Green Buildings								0
8	Rudge et al. 2005 [50]	Fuel Poverty	#							
9	Webb et al. 2013 [51]	Fuel Poverty	#							
10	Sharpe et al. 2015 [52]	Fuel Poverty	#							
11	Poortinga et al. 2017 [53]	Fuel Poverty	+		+/−	+		+/−		
12	Carlton et al. 2019 [54]	Fuel Poverty	−	−						
13	Howden-Chapman et al. 2011 [55]	Fuel Poverty	+		+	+				
14	Howden-Chapman et al. 2007 [56]	Fuel Poverty	+		+					
15	Humphrey et al. 2020 [54]	Fuel Poverty		+						
16	Thomson et al. 2017 [57]	Fuel Poverty			#	#				
17	Ahrentzen et al. 2016 [58]	Fuel Poverty			+	+				
18	Shortt et al. 2007 [59]	Fuel Poverty			+	+				
19	Chapman et al. 2009 [60]	Fuel Poverty			+					
20	Grey et al. 2017 [61]	Fuel Poverty				+		+		
21	Poortinga et al. 2018 [62]	Fuel Poverty				+		+		
22	Pollard et al. 2019 [63]	Fuel Poverty						#		
23	Angelini et al. 2019 [64]	Fuel Poverty							#	
24	Sartini et al. 2018 [65]	Fuel Poverty							+	
25	Peralta et al. 2017 [66]	Fuel Poverty							+/−	
26	Umishio et al. 2019 [67]	Fuel Poverty							#	
27	López-Bueno et al. 2020 [68]	Fuel Poverty							+	
28	Engvall et al. 2003 [69]	Cross sectional								−
29	Smedje et al. 2017 [70]	Cross sectional								+
30	Norback et al. 2014 [71]	Cross sectional		+/−						
31	Wang et al. 2017 [72]	Cross sectional	+							
32	Sharpe et al. 2019 [73]	Cross sectional	+							
33	Sobottka et al. 1996 [74]	Cross sectional								−
34	Bakke et al. 2008 [46]	Cross sectional			#					
35	Kennard et al. 2020 [75]	Cross sectional			#					
36	Wallner et al. 2017 [76]	Longitudinal			+					
37	Somerville et al. 2000 [77]	Intervention	+	+						
38	Barton et al. 2007 [78]	Intervention	+	+						
39	Osman et al. 2010 [79]	Intervention	+							
40	Wilson et al. 2013 [80]	Intervention		+	+			+		
41	Haverinen-Shaughnessy et al. 2018 [81]	Intervention	+					+		
42	Wargocki et al. 2000 [43]	Intervention					+			+
43	Engvall et al. 2005 [82]	Intervention								0
44	Francisco et al. 2017 [83]	Intervention			+					
45	Umishio et al. 2020 [84]	Intervention							+	

Remark: + mean positive result, − means negative result, +/− means mixed results, 0 means no associations, # means fuel poverty issues. SBS: sick building syndrome.

**Table 3 ijerph-19-07393-t003:** Associations between energy factors and any health outcomes among all 45 selected health studies.

No.	References	Energy Efficiency Improvements (at Least Two Measures)	Green Buildings	More Effective Heating	Thermal Insulation	Draught Proofing	Higher Ventilation Rate	Installation of Mechanical Ventilation
1	Garland et al. 2013 [47]		+					
2	Singh et al. 2010 [41]		+					
3	Breysse et al. 2011 [48]		+					
4	Breysse et al. 2015 [49]		+					
5	Hedge et al. 2013 [44]		+					
6	Hedge et al. 2014 [45]		+					
7	Gawande et al. 2020 [42]		+					
8	Poortinga et al. 2017 [61]	+			−			
9	Carlton et al. 2019 [54]						−	
10	Howden-Chapman et al. 2011 [55]			+	+			
11	Howden-Chapman et al. 2007 [56]				+			
12	Humphrey et al. 2020 [85]						+	
13	Ahrentzen et al. 2016 [58]	+						
14	Shortt et al. 2007 [59]			+	+			
15	Chapman et al. 2009 [60]				+			
16	Grey et al. 2017 [61]	+						
17	Poortinga et al. 2018 [62]	+						
18	Sartini et al. 2018 [65]				+			
19	Peralta et al. 2017 [66]				+/−			
20	López-Bueno et al. 2020 [68]			+				
21	Engvall et al. 2003 [69]			+		−		+
22	Smedje et al. 2017 [70]				+			
23	Norback et al. 2014 [71]			+				
24	Wang et al. 2017 [72]						+	
25	Sharpe et al. 2019 [63]					−		
26	Sobottka et al. 1996 [74]					−		
27	Wallner et al. 2017 [76]	+						
28	Somerville et al. 2000 [77]			+				
29	Barton et al. 2007 [78]	+						
30	Osman et al. 2010 [79]	+						
31	Wilson et al. 2013 [80]	+						
32	Haverinen-Shaughnessy et al. 2018 [81]				+			
33	Wargocki et al. 2000 [43]						+	
34	Engvall et al. 2005 [82]						+	
35	Francisco et al. 2017 [83]				+			
36	Umishio et al. 2020 [84]				+			
Positive associations (+)		8	7	6	9		4	1
Negative associations (−)					1	3	1	
Mixed results (+/−)					1			

Remark: + mean positive result, − means negative result, +/− means mixed results.

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
