# Peer review of "A Systematic Review of Associations between Energy Use, Fuel Poverty, Energy Efficiency Improvements and Health"

_ijerph, 2022, doi:10.3390/ijerph19127393_

Round 1

Reviewer 1 Report

Highlight changes in yellow in a next revision, please. No track changes.

I could not find the authors response anywhere.

Because I was extensive before, I will see if my advices were followed or not

Authors changed the title to:

“A Systematic Review of Associations between Energy Use, Fuel Poverty, Energy Efficiency Improvements and Health”

Hence, methodology needs to be extremely detailed in this paper now

This is a long time ago…

“for articles published up to 1st October 2020”

The entire paper would need to be updated…

So general…

“The included studies reported that green buildings can improve 14
health 2

Nt a single reference:

“1. Introduction 26

In modern society, people spend more than 90% of the time in indoor environments 27

and most of the time is spent at home. Energy is needed to heat or cool buildings and 28

energy use in buildings is an important issue in the modern society. The climate change 29

issue, linked to increased greenhouse gases emissions from coal, oil or gas combustion, 30

has increased the demand to save energy in buildings in different parts of the world. Be- 31

cause of this demand, different measures have been applied to increase energy efficiency 32

in buildings in order to create a sustainable built environment which combine a healthy 33

and energy efficient indoor environment. However, since this is a complex issue, scientists 34

from many disciplines as well as stake holders, government officers and other decision 35

makers need to work together”

References?!

“Com- 41

mon energy saving measures include increased thermal insulation, installation of central 42

heating or space heating, draught proofing or installation of a mechanical ventilation with 43

heat recovery system.”

An introduction with 7 references is not acceptable…

Please clarify how it was done. The methodology of a Prisma paper needs to be extremely well detailed.

“Since many search terms, e.g. energy, building, efficient or efficiency are broad search 76

terms, many irrelevant articles were excluded.”

I am sorry to say that the authors have decided to use a repetitive style to describe the results. This is just not done like this. Authors must find a way to make all results relevant without having so much similar tables>

“3. Results 82

The articles were organized into 10 different tables.”

I do not see great interest in having so much tables presented like this. Authors should find a way to connect all content without listing it.

You could add diagrams, figures, etc. Connecting the collected content.

Like this, it’s a sort of separated case studies to each subject without real connection.

Table 11 was in fact most interesting table because the other ones are not that relevant as presented

Is this methodology again?

“This literature review has a number of strengths. The main focus was on epidemio- 373

logical studies, including intervention studies, cross-sectional studies and longitudinal 374

studies. However, exposure studies without any reported health data or health associa- 375

tions were also included if they were identified in this literature search. For each included 376

study, the country of the study, type of study, type of buildings, number of buildings and 377

number of subjects were noted in the review. In exposure studies, extra information on 378

the changes of concentrations of major pollutants was collected. In studies with unex- 379

pected results or negative impacts of energy use and energy saving, explanations of the 380

results reported by the authors were included.”

Read the sentence again to understand how English needs revision. Singular plural?

“The studies included in this review had some limitations.”

I am sorry to say that once again I do not find the review relevant as presented, nor the conclusions which are just too general.

I would advise the authors to go back to my previous review and see if that helps.

Author Response

Comments to the Author:

I could not find the authors response anywhere. Because I was extensive before, I will see if my advices were followed or not. Authors changed the title to: “A Systematic Review of Associations between Energy Use, Fuel Poverty, Energy Efficiency Improvements and Health” Hence, methodology needs to be extremely detailed in this paper now

Answer: we have added three paragraphs clarifying the methodology

Comments to the Author:

This is a long time ago… “for articles published up to 1st October 2020” The entire paper would need to be updated…

Answer: we are very sorry that we cannot update time frame in so short time.

Comments to the Author:

 So general… “The included studies reported that green buildings can improve health.

Answer: The green building concept covers many aspects of a building, and the text limitation does not allow us to explain this. We just write “classified as green buildings”

Comments to the Author:

 Nt a single reference: “Introduction In modern society, people spend more than 90% of the time in indoor environments and most of the time is spent at home. Energy is needed to heat or cool buildings and energy use in buildings is an important issue in the modern society. The climate change issue, linked to increased greenhouse gases emissions from coal, oil or gas combustion, has increased the demand to save energy in buildings in different parts of the world. Because of this demand, different measures have been applied to increase energy efficiency in buildings in order to create a sustainable built environment which combine a healthy and energy efficient indoor environment. However, since this is a complex issue, scientists from many disciplines as well as stake holders, government officers and other decision makers need to work together” References?! “Common energy saving measures include increased thermal insulation, installation of central heating or space heating, draught proofing or installation of a mechanical ventilation with heat recovery system.” An introduction with 7 references is not acceptable…

Answer: we have re-written the introduction and have included references for each sentences as well as have more references in the introduction.

 Comments to the Author:

Please clarify how it was done. The methodology of a Prisma paper needs to be extremely well detailed. “Since many search terms, e.g. energy, building, efficient or efficiency are broad search terms, many irrelevant articles were excluded.”

Answer: we have added three paragraphs clarifying the methodology

Comments to the Author:

 I am sorry to say that the authors have decided to use a repetitive style to describe the results. This is just not done like this. Authors must find a way to make all results relevant without having so much similar tables> “3. Results The articles were organized into 10 different tables.” I do not see great interest in having so much tables presented like this. Authors should find a way to connect all content without listing it. You could add diagrams, figures, etc. Connecting the collected content. Like this, it’s a sort of separated case studies to each subject without real connection.

Answer: we have shorten our results and moved many tables to appendix. Instead, we have added two new tables which summarize results of the studies.

Comments to the Author:

Table 11 was in fact most interesting table because the other ones are not that relevant as presented. Is this methodology again? “This literature review has a number of strengths. The main focus was on epidemiological studies, including intervention studies, cross-sectional studies and longitudinal studies. However, exposure studies without any reported health data or health associations were also included if they were identified in this literature search. For each included study, the country of the study, type of study, type of buildings, number of buildings and number of subjects were noted in the review. In exposure studies, extra information on the changes of concentrations of major pollutants was collected. In studies with unexpected results or negative impacts of energy use and energy saving, explanations of the results reported by the authors were included.”

Answer: we have mentioned this in the methodology part.

 Comments to the Author:

Read the sentence again to understand how English needs revision. Singular plural? “The studies included in this review had some limitations.”

Answer: we have clarified the different types of limitations, an corrected the language.

Comments to the Author:

 I am sorry to say that once again I do not find the review relevant as presented, nor the conclusions which are just too general. I would advise the authors to go back to my previous review and see if that helps.

Answer: we have shorten our results and included new findings in the discussion, trying to be  more specific in the conclusions.

Reviewer 2 Report

I think that although the contribution of this paper to the field is not high, the research is well performed and its results could be useful for certain researchers or other stakeholders.

Author Response

Comments to the Author:

I think that although the contribution of this paper to the field is not high, the research is well performed and its results could be useful for certain researchers or other stakeholders.

Answer: we have shorten our results and included new findings in the discussion, trying to be  more specific in the conclusions.

Reviewer 3 Report

Abstract is clear and well written.

Introduction has clear justification and rationale although it would be helpful to have published literature cited for issues. For example, when saying that energy use in buildings is important in modern society (line 29), support this with several published studies.

The introduction should be expanded to include more detail on energy poverty and the link to poor physical and mental health.

There also needs to be a literature review / background section which includes studies on how improved energy efficiency in residential buildings is linked to health.

The methodology should have some indication of how relevant papers were reviewed e.g. thematic analysis? This is presented in the Results (line 82-93) but could be included in the Methodology.

Results are quite repetitive and don’t add much to material in tables.

Some important fuel poverty studies (section 3.3) are not included e.g. https://doi.org/10.1016/j.enpol.2010.01.037

The risks of radon etc are presented in the discussion. It may be more effective to present the main risks of the common pollutants etc earlier in the review (e.g. Introduction). Ideally it would be useful to have discussion on what radon is, why it would be higher in a well-insulated building, how it impacts on human health and how it can be reduced. Same applies to mould, bacteria and dust mites, chemicals etc.

It would also be useful to know how much radon there is in a home post energy efficiency measures e.g. do concentration levels increase significantly or do levels stay well below dangerous levels?

Recommendations could be developed further with stronger policy focus.

It is unclear why some text is in red font.

Some spelling and grammatical errors throughout (e.g. lines 222, 236 “low-come urban”; lines 289, 300, 350 “may due to”)

Author Response

Comments to the Author:

Abstract is clear and well written.Introduction has clear justification and rationale although it would be helpful to have published literature cited for issues. For example, when saying that energy use in buildings is important in modern society (line 29), support this with several published studies. The introduction should be expanded to include more detail on energy poverty and the link to poor physical and mental health.There also needs to be a literature review / background section which includes studies on how improved energy efficiency in residential buildings is linked to health.

Answer: we have added references in the introduction and we have included two paragraphs in the introduction on energy poverty and energy efficiency in residential buildings.

Comments to the Author:

 The methodology should have some indication of how relevant papers were reviewed e.g. thematic analysis? This is presented in the Results (line 82-93) but could be included in the Methodology.

Answer: we have added three paragraphs clarifying the methodology.

Comments to the Author:

 Results are quite repetitive and don’t add much to material in tables.

Answer: we have shorten our results and moved many tables to appendix. Instead, we have added two new tables which summarize results of the studies.

 Comments to the Author:

Some important fuel poverty studies (section 3.3) are not included e.g. https://doi.org/10.1016/j.enpol.2010.01.037

Answer: This link is for a review article in a journal not included in PubMed. We may have missed some publications not cited in PubMed in our review, especially if they were not about health associations. This is mentioned in the limitations. However, since this review article is interesting, we have now cited it in the introduction part. 

Comments to the Author:

The risks of radon etc are presented in the discussion. It may be more effective to present the main risks of the common pollutants etc earlier in the review (e.g. Introduction). Ideally it would be useful to have discussion on what radon is, why it would be higher in a well-insulated building, how it impacts on human health and how it can be reduced. Same applies to mould, bacteria and dust mites, chemicals etc.

Answer: we have now discussed more on the risks of radon, mould and chemicals in the discussion.

Comments to the Author:

It would also be useful to know how much radon there is in a home post energy efficiency measures e.g. do concentration levels increase significantly or do levels stay well below dangerous levels?

Answer: radon concentration in 6 studies was above 100 Bq/m3 in mean or in rooms. We have included this information in the discussion.

Comments to the Author:

Recommendations could be developed further with stronger policy focus.

Answer: except for winter fuel payment and energy intervention policy, wearable telemetry may be a good choice of solutions. We have included this information in the discussion.

Comments to the Author:

Some spelling and grammatical errors throughout (e.g. lines 222, 236 “low-come urban”; lines 289, 300, 350 “may due to”)

Answer: we have corrected it.

It is unclear why some text is in red font.

Answer: red font marks the places where we make revisions.

Round 2

Reviewer 1 Report

Highlight changes in yellow in a next revision, please. No track changes.

It is acknowledged that the abstract should always start contextualising the need to write this paper with a brief sentence instead of going directly to the objectives.

Authors need to take very care with writing, avoiding duplication words

increased ventilation flow can increase

To me this is not acceptable using extensive parts of the text from the same reference…references.

“[2]. The climate 29

change issue, linked to increased greenhouse gases emissions from coal, oil or gas com- 30

bustion, has increased the demand to save energy in buildings in different parts of the 31

world [2]. Because of this demand, different measures have been applied to increase en- 32

ergy efficiency in buildings in order to create a sustainable built environment which 33

combine a healthy and energy efficient indoor environment [2]. 34

There are there main principles of energy efficiency improvements in buildings: 35

reduced air leakage, reduced heat transfer and reduced energy use [2,3]. In practice, 36

common energy saving measures in buildings include increased thermal insulation, in- 37

stallation of central heating or space heating, draught proofing or installation of heat 38

recovery system [2,3]. Reduced energy use can reduce emissions and fuel cost so that 39

reduce exposure to emissions [2,3]. Reduced heat transfer can increase indoor tempera- 40

ture and reduce relative humidity and risk of mould [2,3]. In contrast, reduced air leak- 41

age can increase relative humidity and risk of mould [2,3].”

Search strings need to be included in the methods.

The reviewer uses time wisely to make extensive comments helping the authors to have a relevant test, and the authors answer with such brief responses that says nothing.

Comments to the Author:

Table 11 was in fact most interesting table because the other ones are not that relevant as presented. Is this methodology again? “This literature review has a number of strengths. The main focus was on epidemiological studies, including intervention studies, cross-sectional studies and longitudinal studies. However, exposure studies without any reported health data or health associations were also included if they were identified in this literature search. For each included study, the country of the study, type of study, type of buildings, number of buildings and number of subjects were noted in the review. In exposure studies, extra information on the changes of concentrations of major pollutants was collected. In studies with unexpected results or negative impacts of energy use and energy saving, explanations of the results reported by the authors were included.”

Answer: we have mentioned this in the methodology part.

Authors must find a way to present the results in a relevant way. There is no interest in having lists of topics to be addressed during the text. That is why a software such as VOSviewer is so important because it allows us to see the connections between the subjects in a different way. Please consider includes schemes, rather than tables and extensive text.

Addresee all italics in variables

dom (n=2), Norway (n=1), Germany (n=1).

Tables with results are not meant to the to the discussion section but to the results section

Table 3: Associations between energy factors and health outcomes among all 45 selected health 290
studies.

Because this is a review, the discussion section would have to be much more extensive. The turnaround is to emerge it with the results section. Think about it.

Again, because this is a review, the conclusion section would have to be more detailed because the authors opted to separate the results in two very different sections and they are not really all addressed here. These are very general conclusions.

I hope the authors understand that aim of this is to expand the interest of the paper.

Author Response

It is acknowledged that the abstract should always start contextualising the need to write this paper with a brief sentence instead of going directly to the objectives.

Answer: We have started the abstract with a brief sentence to describe the need: Energy use in buildings can influence indoor environment.. Since the limitation of words (200 words) for abstract, many sentences may have to be very short.

Authors need to take very care with writing, avoiding duplication words“increased ventilation flow can increase”

Answer: We have now changed increased ventilation flow can increase to higher ventilation flow can increase and have removed some duplication in conclusion.

To me this is not acceptable using extensive parts of the text from the same reference…references. “[2]. The climate change issue, linked to increased greenhouse gases emissions from coal, oil or gas com bustion, has increased the demand to save energy in buildings in different parts of the world [2]. Because of this demand, different measures have been applied to increase energy efficiency in buildings in order to create a sustainable built environment which combine a healthy and energy efficient indoor environment [2]. There are there main principles of energy efficiency improvements in buildings: reduced air leakage, reduced heat transfer and reduced energy use [2,3]. In practice, common energy saving measures in buildings include increased thermal insulation, installation of central heating or space heating, draught proofing or installation of heat recovery system [2,3]. Reduced energy use can reduce emissions and fuel cost so that reduce exposure to emissions [2,3]. Reduced heat transfer can increase indoor temperature and reduce relative humidity and risk of mould [2,3]. In contrast, reduced air leakage can increase relative humidity and risk of mould [2,3].”

Answer: We agree that it is not a good idea to cite many sentences from the same reference. We have updated texts without using the same reference for many sentences. In the introduction, we have described energy use in the beginning, then shifted it to the need to save energy and the need to build energy efficiency buildings, green buildings, healthy buildings but the need to consider fuel poverty issues.

Search strings need to be included in the methods.

Answer: We have added more details on search strings: Any medical search term mean OR between each search term. Any energy or building related search term means OR between each search term. Combined means AND between the two groups of search terms. There are other ways to describe search strings. However, some of them are very hard for readers to understand, especially including many combinations. Thus, we think this way is better.

The reviewer uses time wisely to make extensive comments helping the authors to have a relevant test, and the authors answer with such brief responses that says nothing.“Comments to the Author: Table 11 was in fact most interesting table because the other ones are not that relevant as presented. Is this methodology again? “This literature review has a number of strengths. The main focus was on epidemiological studies, including intervention studies, cross-sectional studies and longitudinal studies. However, exposure studies without any reported health data or health associations were also included if they were identified in this literature search. For each included study, the country of the study, type of study, type of buildings, number of buildings and number of subjects were noted in the review. In exposure studies, extra information on the changes of concentrations of major pollutants was collected. In studies with unexpected results or negative impacts of energy use and energy saving, explanations of the results reported by the authors were included. Answer: we have mentioned this in the methodology part.”

Answer: This part is about the strength of our methodology and we think it belongs to discussion. However, we have now described what kind of data we collected in the methods. We are very sorry that we previously answered very short. We always try to follow suggestions by the reviewers, but some comments can be a bit unclear and written in general terms.

Authors must find a way to present the results in a relevant way. There is no interest in having lists of topics to be addressed during the text. That is why a software such as VOSviewer is so important because it allows us to see the connections between the subjects in a different way. Please consider includes schemes, rather than tables and extensive text.

Answer: We have not used VOSviewer before, but we tested it now. It can be good for researcher to gather ideas on frequent keywords in a field or research groups. However, it seems not relevant to use in this review article. We have followed a much more common style as used in other reviews in the field of environmental epidemiology. However, we have modified many different parts of the review and hope that reviewer can be more satisfied with the current version.

Address all italics in variables“dom (n=2), Norway (n=1), Germany (n=1).”

Answer: we have now addressed all italics in variables.

Tables with results are not meant to the to the discussion section but to the results section. “Table 3: Associations between energy factors and health outcomes among all 45 selected health studies.”

Answer: we have moved this table 3 to result part.

Because this is a review, the discussion section would have to be much more extensive. The turnaround is to emerge it with the results section. Think about it.

Answer: We have now expanded discussion parts which both editor and other reviewer asked for.

Again, because this is a review, the conclusion section would have to be more detailed because the authors opted to separate the results in two very different sections and they are not really all addressed here. These are very general conclusions.

Answer: conclusion section has been expanded, especially regarding suggestions of future research (asked for by the editor) and added some specific suggestions on technical solutions. We have separated the results in two different sections, this is because some researchers may have currently focused on indoor environment and other researchers may have currently focused on health.

I hope the authors understand that aim of this is to expand the interest of the paper.

Answer: we thank the reviewer for valuable comments on our review. Hopefully, it has been improved now.

This manuscript is a resubmission of an earlier submission. The following is a list of the peer review reports and author responses from that submission.

Round 1

Reviewer 1 Report

This paper reviews the main and more recent scientific works studiying the relationships between energy efficiency and health in buildings. It covers an issue that is relevant for the scientific community and for society in general: adressing energy conservation in buildings while reducing the associated health risks for their users. It provides a rather good insigh on the matter through a rigurous list of works. However, it presents some weaknesses, which in my opinion are points that authors should improve before the paper is accepted for publication: i) the gap in the literature is not sufficiently identified or justified; ii) the discussion derived from the cited papers is, in general, poor and repetitive; that is, a more in deep analysis of certain references is desirable.

Reviewer 2 Report

Thank you for the opportunity to review this review article, which follows numerous reviews in the field (at least 11, or perhaps 12?) but aims to take a broader and more comprehensive perspective, including all health aspects of energy efficiencies/healthy buildings. The authors reviewed nearly 5,000 articles and summarized 68 articles, in an organized fashion.

The review lacked quantitative summarization (e.g., meta-analysis), and I have no additional comments which would help to strengthen the work. This review would be a useful starting place for someone wanting to understand the literature in this field.

Reviewer 3 Report

Highlight changes in yellow in a next revision, please. No track changes.

Start abstract with brief contextualization, please

Do not use “our” or similar

Clarify: “on …”: “The exposure studies”

Define abbreviations at first sue… “TVOC, CO2, CO”

Cannot find relevant findings: “

 Many studies have used a combination of energy saving measures. In future studies, 21 more focus should be on evaluating health impacts of single energy efficiency improvement 22 measures. Moreover, more prospective health studies on energy efficiency improvements are 23 needed.”

Where is the methodology described, so important in a review, time frame, criteria, screening, etc…

Compare to abstract:

“This literature review has included all types of health aspects of energy 77 use, energy saving and energy efficiency in buildings. The main aim is to summarize the 78 current knowledge on how energy saving measures and energy use influence our health. 79 The second aim is to collect knowledge on how energy saving measures and energy use 80 influence exposure in the indoor environment. The third aim is to gather knowledge on 81 types of energy saving or energy use that should be promoted from a health perspective.”

Mention needs to be in the abstract: “This review was performed and followed the guidelines by the Preferred Reporting 84 Items for Systematic reviews and Meta-Analyses (PRISMA) statement [20].”

Address italics in the figure “n”: “Figure 1. PRISMA flow diagram of literature research.”

Add time frame to caption

I would avoid this continuous heading style without further explanation:

3. Results 106

3.1. Exposure studies 107

3.1.1. Radon”

Write differently:

“Some studies reported”

“However, some studies reported”

Avoid starting like this every time:

“3.1.1. Radon 108

Table 1 lists and summarizes studies on how energy saving and energy use influence 109 indoor radon exposure.”

“3.1.2. Mould, Bacteria and House Dust Mites 135

Table 2 lists and summarizes studies on how energy saving and energy use influence 136 indoor exposure to mould, bacteria and house dust mites.”

And goes on…

And on

And on…

(…)

3.6. Intervention Health Studies 315

Table 10 lists and summarizes”

Not possible

This is not the correct way to start, I suggest authors o read more:

4. Discussion 341

In total, 68 relevant peer-reviewed epidemiological or exposure studies were in-342 cluded.”

There is no real discussion, so important in reviews…

Again, in a review, it is crucial to demonstrate something new, so the discussion and conclusions are crucial.

5. Conclusions

To include:

Brief contextualization and methodology, main findings and practical implications

What really resulted from this review?

Authors limit to repeat the abstract, unclear…

Abstract:

“ However, inadequate ventilation, heating, air tightness may have a negative effects on health. En-19 ergy efficiency improvements combined with installation of a mechanical ventilation system should 20 be promoted. Many studies have used a combination of energy saving measures. In future studies, 21 more focus should be on evaluating health impacts of single energy efficiency improvement 22 measures. Moreover, more prospective health studies on energy efficiency improvements are 23 needed.”

Conclusions

“However, 413 increased ventilation flow can increase the indoor exposure to outdoor air pollutant in 414 areas with high levels of outdoor air pollution. Since most of studies were on residential 415 buildings, more future studies on energy use and health in office, school and hospital 416 buildings should be conducted.”

Tables cannot be at the end and reference number must be included after authors names, in every case (single column)

Tables are not relevant to me, since authors limited to collected data in each case, no real innovation